# Diabetic control and compliance using glycated haemoglobin (HbA1C) testing guidelines in public healthcare facilities of Gauteng province, South Africa

**Ngalulawa Kone[1,2]⊙\*, Naseem Cassim[2,3]⊙, Innocent Maposa[4,5]⊙, Jaya Anna George[1,2]⊙**

**1** Department of Chemical Pathology, Faculty of Health Sciences, University of Witwatersrand, Johannesburg, South Africa, **2** National Health Laboratory Service (NHLS), Johannesburg, South Africa, **3** Department of Molecular Medicine and Haematology, Faculty of Health Sciences, University of Witwatersrand, Johannesburg, South Africa, **4** Division of Epidemiology and Biostatistics, School of Public Health, Faculty of Health Sciences, University of Witwatersrand, Johannesburg, South Africa, **5** Division of Epidemiology and Biostatistics, Department of Global Health, Faculty of Medicine and Health Sciences, Stellenbosch University, Stellenbosch, South Africa

⊙ These authors contributed equally to this work.

\* ngalula.kone@nhls.ac.za

**Data Availability Statement:** The data is available from the National Health Laboratory Service Corporate Data Warehouse, subject to an

## Abstract

### Objective

This study aimed at evaluating diabetic control and compliance with testing guidelines, across healthcare facilities of Gauteng Province, South Africa, as well as factors associated with time to achieve control. South Africa's estimated total unmet need for care for patients with type 2 diabetes mellitus is 80%.

### Research design, methods and findings

The data of 511 781 patients were longitudinally evaluated. Results were reported by year, age category, race, sex, facility and test types. HbA1C of ≤7% was reported as normal, >7 - ≤9% as poor control and >9% as very poor control. The chi-squared test was used to assess the association between a first-ever HbA1C status and variables listed above. The Kaplan-Meier analysis was used to assess probability of attaining control among those who started with out-of-control HbA1C. The extended Cox regression model assessed the association between time to attaining HbA1C control from date of treatment initiation and several covariates. We reported hazard ratios, 95% confidence intervals and p-values. Data is reported for 511 781 patients with 705 597 laboratory results. Poorly controlled patients constituted 51.5%, with 29.6% classified as very poor control. Most poorly controlled patients had only one test over the entire study period. Amongst those who started with poor control status and had at least two follow-up measurements, the likelihood of achieving good control was higher in males (adjusted Hazard Ratio (aHR) = 1.16; 95% CI:1.12–1.20; p<0.001) and in those attending care at hospitals (aHR = 1.99; 95% CI:1.92–2.06; p<0.001).

application to the Academic Affairs and Research Office. This application can be made using the online system (https://aarms.nhls.ac.za/).

**Funding:** The authors received no specific funding for this work.

**Competing interests:** The authors have declared that no competing interests exist.

## Conclusion

This study highlights poor adherence to guidelines for diabetes monitoring.

## Introduction

Diabetes mellitus is an escalating public health crisis and is among the top ten leading causes of death worldwide [1]. South Africa has the second highest number of people living with type 2 diabetes mellitus (T2DM) in sub-Saharan Africa and, T2DM is currently ranked second among the top ten leading natural causes of death [2–5]. A systematic review published in 2021, estimated that the pooled prevalence of T2DM in South Africans aged 25 years and older was 15.25% (95% CI: 11.07–19.95%) [3]. According to the 2020 data from the World Health Organization (WHO), deaths from diabetes mellitus account for 6.64% of all deaths in South Africa (compared to 1,52% in Nigeria, 2,45% in the United States, and 3,2% in India), placing the country in position 16 globally [6]. The International Diabetes Federation (IDF) projected that diabetes will affect 42 340 people aged 20 to 79 in 2021, rising to 54 452 by 2030, representing a comparable age-adjusted prevalence of 12,2%, while the percentage of those without a diagnosis of the condition was 45.4% [7].

Responding to the increasing global burden of diabetes, the WHO has launched the Global Diabetes Compact (GDC) [8]. For the first time ever, WHO member states have supported the creation of global targets for diabetes, as part of recommendations to strengthen and monitor diabetes responses within national non-communicable disease (NCD) programmes. Two of the targets are to ensure 80% of people living with diabetes are diagnosed, and that 80% of those diagnosed have good glycaemic control [8]. Targets for South Africa set in 2013, called for a 30% increase in testing patients with controlled diabetes by 2020, using HbA1C [9, 10].

HbA1C should be tested at a patient's first visit, and then regularly as part of continuing care, to aid in therapeutic decisions. The Society for Endocrinology Metabolism and Diabetes of South Africa (SEMDSA) recommends that HbA1C be measured six monthly in stable controlled diabetic patients, and every three months in those who are not reaching their target goals or who have had therapy changed [10]. Similar guidelines have been produced by the American Diabetes Association (ADA) [11]. In a local study in the province of KwaZulu-Natal, HbA1c was measured in 29.2% of patients once and 13.2% of patients twice in the past year. Evaluation of selected records demonstrated compliance with the SEMDSA guidelines in only 4.2% of patients [12]. While there is some evidence to support the use of HbA1C for the diagnosis of diabetes, this has not been validated in African populations [13].

The aim of this study was to assess diabetic control and compliance with local testing guidelines over a four-year period (January 2015- December 2018), across primary healthcare facilities and hospitals in the Gauteng province, South Africa. Factors associated with time to HbA1C control among patients who were poorly controlled when they started in care, were also explored.

## Research design, materials and methods

### Ethics statement

Ethical clearance was obtained in writing from the Human Research Ethics Committee (HREC)(Medical), at the University of the Witwatersrand (M1911163).

## Study design

The longitudinal study design was used to describe HbA1C results extracted from the Central Data Warehouse (CDW) of the National Laboratory Health Services (NHLS), for all patients tested between January 2015 and December 2018, in the public sector of Gauteng healthcare facilities. The CDW houses all laboratory results for public sector patients in South Africa.

## Study site & population

After obtaining ethics approval, all HbA1C data for the province was retrieved. The study included all male and female patients, aged ≥18 years, tested in healthcare facilities (clinics) and hospitals in Gauteng, over the study period.

## Inclusion criteria

The inclusion criteria were: (i) reviewed data within the study period, (ii) results for CDW unique patient identifier of -1 (unmatched) or -2 (study participants), (iii) an age of ≥18 years and (iv) an HbA1C of ≥3.5 and ≤20.

## Exclusion criteria

Patients <18 years (children), those with missing unique identifiers, and with HbA1C levels <3.5 and >20% (out of equipment analytical reference range), were excluded from the study (reference package insert).

## Definitions

We defined tests as either "too soon" for those with an interval between requests of <6 months in patients with an initial HbA1C value of <7.0% and < 3 months in patients with an initial HbA1C of >7.0%, or "too late" for those requested >12 months after the previous test in patients with an initial HbA1C value of <7.0% and > 6 months later in patients with an initial HbA1C of > 7.0%. The previous HbA1C value was used to define the appropriate testing frequency for the next request. An HbA1C >7–9% was categorised as poor control, and > 9% as very poor control [14].

## Data extract

The data extract included the following variables: (i) episode number, (ii) unique patient identifier, (iii) date reviewed, (iv) age (in years) (v) race group, (iv) sex, (v) facility description, (vi) clinical diagnosis, (vii) test (laboratory/Point of Care (POC)), (viii) HbA1C result (text and numeric) and (ix) tested date. We used the unique patient identifier that is generated by a probabilistic matching algorithm developed by the CDW that includes fuzzy logic matching. This algorithm uses a set of demographic variables to de-duplicate patients. Bassett *et al* have shown that using the CDW unique patient identifier, they were able to identify a cohort of HIV patients transferred from a hospital to Primary Health Care (PHC) facilities with 90% accuracy [15].

## Data preparation

We used the hot-deck imputation method described in a local study which is administered by the National Cancer Registry, due to the paucity of race group data. We categorised the data, using age in years, as follows: (i) 18–30, (ii) 31–50, (iii) >50 and (iv) Unknown for missing values. The HbA1C values were captured as string and numerical values. String values were converted to numbers using a lookup table, e.g., '<3.6' and '>18.5' converted to 3.5 and 18.6,

respectively. In this way all HbA1C results were converted to numerical values and categorised as follows: (i) ≤7, (ii) >7 - ≤9 and (iii) >9%. We used the facility description to assign the facility type as follows: (i) hospital, (ii) PHC facility and (iii) Other. For example, the facility description 'BERTHA GXOWA HOSPITAL' was assigned the "hospital" facility type. The unique patient identifier has been described locally to generate a sequence number for chronologically sorted patient data [16]. We determined the difference between consecutive results for a unique patient using the DATEDIF function (calculated in months). In addition, the cumulative date difference between consecutive results was added up (last result reported the total date difference from the first result). The clinical diagnosis captured on the request form was coded using a lookup table, e.g., '? DIABETES MELLITUS/HPT' was assigned as confirmed diabetes mellitus.

### Statistical analysis

The data was prepared using Microsoft Excel (Redmond, CA, USA) and analysed using SAS (Version 9.4, Cary, NC, USA) and Stata SE (Version 17, College Station, TX, USA).

We reported the number of samples by year, age category, race, sex, facility type and test type. The number of samples with an HbA1C category of ≤7%, >7 - ≤9% and >9% were also reported. This analysis was repeated for first-ever HbA1C, with a sequence number of one, i.e., index HbA1C defined as the first test during the four-year study period. The chi-squared test was used to assess the association between a first-ever HbA1C category and year, age category, race group, sex, facility type and test type. Based on each HbA1C result, we determined whether the next test was too soon, too early or, in compliance with local guidelines [10]. An HbA1C of ≤7 and >7 should receive a follow up test at 6 and 3 months, respectively [10]. We used the difference between consecutive results for the same patients to assign compliance with guidelines from the second result onwards. We assessed what percentage of the overall testing was compliant with guidelines. This was repeated by sex, age category and facility type. A descriptive analysis of testing by sequence number was conducted (1 to 11 and categorised the remaining as ≥12). We reported the number of samples, median HbA1C with interquartile range, HbA1C categories and median follow-up period (in months). We compared guidelines compliance between the first and second HbA1C, with the categories reported for the second result. We reported the number of samples and median follow-up period. Furthermore, an analysis of a subsample of patients who entered care with a first-ever-out-of-control HbA1c with at least two follow-up measurements was performed. To determine factors associated with time to control, the extended Cox regression model was used to assess association between time to attaining HbA1C control with demographic factors. We reported adjusted hazard ratios, 95% confidence intervals and p-values. We also conducted a Kaplan-Meier analysis for age category, race group, sex and facility type to determine the cumulative likelihood of attaining HbA1C control given the months from treatment initiation with a first-ever out of control HbA1C.

## Results

We extracted 717 360 test results, after which 11 763 samples were excluded (1.6%). This was largely made up of those with an age <18 years (n = 9 751). Data is reported for 705 597 results and 511 781 patients following de-duplication (Fig 1).

### Descriptive analysis

Overall, there were 348 962 (49.5%), 147 741 (20.9%) and 208 894 (29.6%) results with an HA1C <7%, >7-≤9% and >9% respectively (Table 1). Test samples increased from 131 529 in 2015 to 213 257 in 2018 (Table 1).

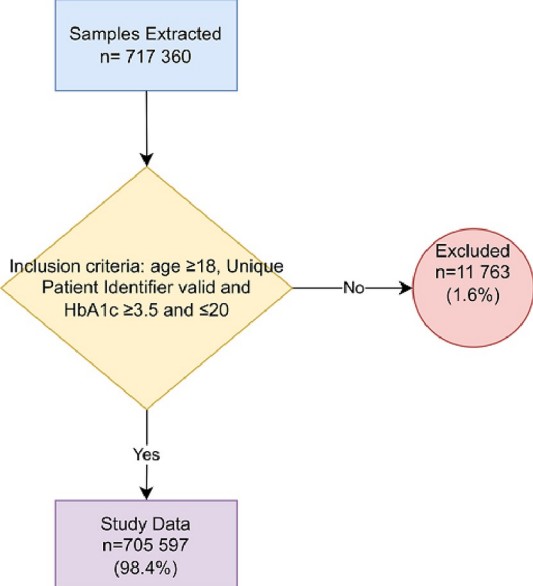

The inclusion criteria included an age ≥18, a valid unique patient identifier and an HbA1c ≥3.5 and ≤20.

**Fig 1. Flow chart depicting the application of the inclusion criteria on the data extract.** The inclusion criteria included an age ≥18, a valid unique patient identifier and an HbA1C ≥3.5 and ≤20.

The majority of testing was performed for those aged >50 years and in Black African people (60.0% and 72.2% respectively). More females were tested compared to males (60.7% vs 36.1%). Most tests were performed on patients attending hospitals (59.5%) in contrast to 40.3% for PHC facilities. Only 3.8% of testing was performed on POC instruments (Table 1).

Overall, 356 635 (51.5%) of test results were >7%, with 29.6% classified as very poor control (HbA1C >9%). The proportion of patients with very poor control was not very different across age groups. Total proportions of those with HbA1C >7% increased from 38% of those between the ages of 18 to 30 years to 52.5% of patients over the age of 50. Proportionately, results from those attending hospitals showed better control (56.0%) compared to those attending PHC facilities (39.8%). There were proportionately more POC tests done on poorly controlled patients (Table 1).

There was a steady increase in numbers of patients with a first-ever HbA1C, increasing from 115 372 in 2015 to 140 718 in 2018, which is an 22% increase (Table 2). Most of this testing was done on those over 50 years (58%), in females (61.0%) and for Black African people (72.9%). The majority of first-ever HbA1C testing was laboratory-based (98.7%) with only 1.3% patients accessing POC.

Forty-six percent of all those who had a first test had an HbA1C >7%, and overall, 28% were >9%. By age category, those between 18–30 years had the highest proportion of HbA1C tests ≤7% (68.7%). An HbA1C >7% was reported for 44.1% and 49.3% for the 31–50 and >50 age categories respectively. White patients, those tested in hospitals and those who had a laboratory test as compared to POC test had higher proportions of HbA1C of ≤7% (65%, 61.5% and 53.9% respectively). Black African patients displayed the highest percentage of poor control (48.1%). By sex, males and females showed almost equal proportions of poor control (45.4% vs 46.8%).

**Table 1. Descriptive analysis of HbA1C testing between 2015 and December 2018 in the Gauteng province, South Africa.**

| Category | Overall n (% Total) | HbA1C category n (%) | | | |
|---|---|---|---|---|---|
| | | Normal Control | Poor Control | | |
| | | ≤7% | >7 - ≤9% | >9%. | Total >7% |
| **Overall** | 705 597 (100.0) | 348 962 (49.5) | 147 741 (20.9) | 208 894 (29.6) | 356 635 (51.5) |
| **Year** | | | | | |
| 2015 | 131 529 (18.6) | 60 394 (45.9) | 28 591 (21.7) | 42 544 (32.3) | 71 135 (54.1) |
| 2016 | 170 936 (24.2) | 85 729 (50.2) | 35 555 (20.8) | 49 652 (29.0) | 85 207 (49.8) |
| 2017 | 189 875 (26.9) | 96 094 (50.6) | 39 205 (20.6) | 54 576 (28.7) | 93 781 (49.4) |
| 2018 | 213 257 (30.2) | 106 745 (50.1) | 44 390 (20.8) | 62 122 (29.1) | 106 512 (49.9) |
| **Age category** | 705 597 (100.0) | 348 962 (49.5) | 147 741 (20.9) | 208 894 (29.6) | 356 635 (51.5) |
| 18–30 | 41 870 (5.9) | 25 977 (62.0) | 3 936 (9.4) | 11 957 (28.6) | 15 893 (38.0) |
| 31–50 | 183 872 (26.1) | 95 223 (51.8) | 30 001 (16.3) | 58 648 (31.9) | 88 649 (48.2) |
| >50 | 423 085 (60.0) | 201 150 (47.5) | 102 094 (24.1) | 119 841 (28.3) | 221 935 (52.5) |
| Unknown | 56 770 (8.0) | 26 612 (46.9) | 11 710 (20.6) | 18 448 (32.5) | 30 158 (53.1) |
| **Race group** | 705 597 (100.0) | 348 962 (49.5) | 147 741 (20.9) | 208 894 (29.6) | 356 635 (50.5) |
| Indian/Asian | 21 896 (3.1) | 10 686 (48.8) | 5 738 (26.2) | 5 472 (25.0) | 11 210 (51.2) |
| Black African | 509 232 (72.2) | 242 887 (47.7) | 106 732 (21.0) | 159 613 (31.3) | 266 345 (52.3) |
| Coloured | 33 941 (4.8) | 17 858 (52.6) | 7 379 (21.7) | 8 704 (25.6) | 16 083 (47.4) |
| White | 74 234 (10.5) | 43 970 (59.2) | 14 656 (19.7) | 15 608 (21.0) | 30 264 (40.8) |
| Unknown | 66 294 (9.4) | 33 561 (50.6) | 13 236 (20.0) | 19 497 (29.4) | 32 733 (49.4) |
| **Sex** | 705 597 (100.0) | 348 962 (49.5) | 147 741 (20.9) | 208 894 (29.6) | 356 635 (50.5) |
| Female | 428 276 (60.7) | 211 214 (49.3) | 91 432 (21.3) | 125 630 (29.3) | 217 062 (50.7) |
| Male | 254 966 (36.1) | 129 922 (51.0) | 50 568 (19.8) | 74 476 (29.2) | 125 044 (49.0) |
| Unknown | 22 355 (3.2) | 7 826 (35.0) | 5 741 (25.7) | 8 788 (39.3) | 14 529 (65.0) |
| **Facility Type** | 705 597 (100.0) | 348 962 (49.5) | 147 741 (20.9) | 208 894 (29.6) | 356 635 (50.5) |
| PHC Facility | 284 428 (40.3) | 113 218 (39.8) | 73 025 (25.7) | 98 185 (34.5) | 171 210 (60.2) |
| Hospital | 419 664 (59.5) | 234 914 (56.0) | 74 374 (17.7) | 110 376 (26.3) | 184 750 (44.0) |
| Other | 1 505 (0.2) | 830 (55.1) | 342 (22.7) | 333 (22.1) | 675 (44.9) |
| **Test Type** | 705 597 (100.0) | 348 962 (49.5) | 147 741 (20.9) | 208 894 (29.6) | 356 635 (50.5) |
| Laboratory | 678 513 (96.2) | 342 036 (50.4) | 138 953 (20.5) | 197 524 (29.1) | 336 477 (49.6) |
| Point of Care | 27 084 (3.8) | 6 926 (25.6) | 8 788 (32.4) | 11 370 (42.0) | 20 158 (74.4) |

Data is reported for samples submitted to the National Health Laboratory Service (NHLS) site. The overall number of samples is reported as well as for the following HbA1C categories: (i) ≤7, (ii) >7 - ≤9 and (iii) >9%. Poor control was defined as a HbA1C >7%. PHC: Primary Health Care.

## Sequence number analysis

Most patients had only one test over the study period (511 781 (73.53%)), and of these patients, 94 248 (18.4%) were poorly controlled, and even more patients, (143 374 (28%)), very poorly controlled (Table 3). The median HbA1C ranged from 6.8% (5.8–9.5) in those who had one test, to 9.0% (7.4–10.9) in those with ≥12 tests. Follow up testing did not change with level of control (Table 3).

## Guidelines compliance analysis

Table 4 shows the proportions of patients that met testing guidelines according to the level of HbA1C. A median test interval of 11 months was observed over the study period with those with HbA1C >7-≤9% and >9% (82.9% and 82.1% respectively).

Fig 2 illustrates the compliance to guidelines classification with only 5% of patients overall meeting the SEMDSA testing interval guidelines (Fig 2A). Sixty-two percent had their testing

**Table 2. Descriptive analysis of the first-ever HbA1C testing between 2015 and December 2018 in the Gauteng province, South Africa.**

| Category | Overall | | Poor Control | | | | p-value |
|---|---|---|---|---|---|---|---|
| | n (% Total) | Normal Control | | | | | |
| | | ≤7% | >7 - ≤9% | >9% | Total >7% | | |
| Overall | 511 781 (100.0) | 274 159 (53.6) | 94 248 (18.4) | 143 374 (28.0) | 237 622 (46.4) | | |
| Year | | | | | | | |
| 2015 | 115 372 (22.5) | 54 750 (47.5) | 23 880 (20.7) | 36 742 (31.8) | 60 622 (52.5) | | |
| 2016 | 126 202 (24.7) | 68 713 (54.4) | 22 946 (18.2) | 34 543 (27.4) | 57 489 (45.6) | | |
| 2017 | 129 489 (25.3) | 72 346 (55.9) | 22 711 (17.5) | 34 432 (26.6) | 57 143 (44.1) | | |
| 2018 | 140 718 (27.5) | 78 350 (55.7) | 24 711 (17.6) | 37 657 (26.8) | 62 368 (44.3) | | $< .0001$ |
| Age category | 511 781 (100.0) | 274 159 (53.6) | 94 248 (18.4) | 143 374 (28.0) | 237 622 (46.4) | | |
| 18–30 | 33 403 (6.5) | 22 936 (68.7) | 2 450 (7.3) | 8 017 (24.0) | 10 467 (31.3) | | |
| 31–50 | 140 897 (27.5) | 78 790 (55.9) | 20 088 (14.3) | 42 019 (29.8) | 62 107 (44.1) | | |
| >50 | 297 046 (58.0) | 150 582 (50.7) | 64 719 (21.8) | 81 745 (27.5) | 146 464 (49.3) | | |
| Unknown | 40 435 (7.9) | 21 851 (54.0) | 6 991 (17.3) | 11 593 (28.7) | 18 584 (46.0) | | $< .0001$ |
| Race group | 511 781 (100.0) | 274 159 (53.6) | 94 248 (18.4) | 143 374 (28.0) | 237 622 (46.4) | | |
| Indian/Asian | 13 342 (2.6) | 7 157 (53.6) | 3 019 (22.6) | 3 166 (23.7) | 6 185 (46.4) | | |
| Black African | 372 942 (72.9) | 193 457 (51.9) | 69 154 (18.5) | 110 331 (29.6) | 179 485 (48.1) | | |
| Coloured | 21 948 (4.3) | 12 897 (58.8) | 3 959 (18.0) | 5 092 (23.2) | 9 051 (41.2) | | |
| White | 48 694 (9.5) | 31 661 (65.0) | 7 969 (16.4) | 9 064 (18.6) | 17 033 (35.0) | | |
| Unknown | 54 855 (10.7) | 28 987 (52.8) | 10 147 (18.5) | 15 721 (28.7) | 25 868 (47.2) | | |
| Sex | 511 781 (100.0) | 274 159 (53.6) | 94 248 (18.4) | 143 374 (28.0) | 237 622 (46.4) | | |
| Female | 312 254 (61.0) | 166 091 (53.2) | 59 274 (19.0) | 86 889 (27.8) | 146 163 (46.8) | | |
| Male | 188 290 (36.8) | 102 723 (54.6) | 32 653 (17.3) | 52 914 (28.1) | 85 567 (45.4) | | |
| Unknown | 11 237 (2.2) | 5 345 (47.6) | 2 321 (20.7) | 3 571 (31.8) | 5 892 (52.4) | | $< .0001$ |
| Facility Type | 511 781 (100.0) | 274 159 (53.6) | 94 248 (18.4) | 143 374 (28.0) | 237 622 (46.4) | | |
| PHC Facility | 214 118 (41.8) | 91 094 (42.5) | 51 453 (24.0) | 71 571 (33.4) | 123 024 (57.5) | | |
| Hospital | 296 499 (57.9) | 182 379 (61.5) | 42 561 (14.4) | 71 559 (24.1) | 114 120 (38.5) | | |
| Other | 1 164 (0.2) | 686 (58.9) | 234 (20.1) | 244 (21.0) | 478 (41.1) | | $< .0001$ |
| Test Type | 511 781 (100.0) | 274 159 (53.6) | 94 248 (18.4) | 143 374 (28.0) | 237 622 (46.4) | | |
| Laboratory | 505 012 (98.7) | 272 296 (53.9) | 92 095 (18.2) | 140 621 (27.8) | 232 716 (46.1) | | |
| **Point of Care** | **6 769 (1.3)** | **1 863 (27.5)** | **2 153 (31.8)** | **2 753 (40.7)** | **4 906 (72.5)** | | $< .0001$ |

Data is reported for samples submitted to a National Health Laboratory Service (NHLS) site. The overall number of patients with a first-ever HbA1C is categorised as follows: (i) ≤7, (ii) >7 - ≤9 and (iii) >9%. The chi-squared test was used to assess whether an association between HbA1C categories and year, age category, race group, sex, facility type and test type. Poor control was defined as a HbA1C >7%. PHC: Primary Health Care.

performed too late, and this was seen for both males and females (Fig 2B). By age category, there were proportionately more people between the ages of 31–50 years who had testing late, compared to those aged > 50 years (66.0% vs. 62.5% respectively). By facility type, more people attending PHC had testing performed later, compared to those attending hospital (69.3 vs 58.7% respectively).

Fig 3 shows the rates for achieving control among patients who entered into care with out-of-control HbA1C. Patients over 50 years (Fig 3A), white patients (Fig 3B), males (Fig 3C), as well as those who were attending hospitals (Fig 3D) had a higher rate of achieving HbA1C control over time.

Amongst those who started with poor control and had at least two follow-up measurements, the likelihood of achieving good control was higher in males (aHR = 1.16; 95%

**Table 3. Descriptive analysis of the sequence number for HbA1C testing between 2015 and December 2018 in the Gauteng province, South Africa.** A sequence number of ≥12 has been grouped.

| Seq | Number of samples n (%) | Median HbA1C (IQR) | HbA1C category n (%) | | | Median follow-up period months (months) | | |
|---|---|---|---|---|---|---|---|---|
| | | | Good Control | Poor Control | | Good Control | Poor Control | |
| | | | ≤7% | >7 - ≤9% | >9% | ≤7% | >7–9% | >9% |
| 1 | 511 781 (73.53) | 6.8 (5.8–9.5) | 274 159 (53.6) | 94 248 (18.4) | 143 374 (28.0) | | | |
| 2 | 103 331 (14.64) | 7.5 (6.2–9.8) | 44 702 (43.3) | 26 134 (25.3) | 32 495 (31.4) | 10.0 | 11.0 | 11.0 |
| 3 | 41 566 (5.89) | 7.8 (6.4–9.9) | 15 651 (37.7) | 11 791 (28.4) | 14 124 (34.0) | 8.0 | 8.0 | 8.0 |
| 4 | 19 542 (2.77) | 8.0 (6.6–10) | 6 628 (33.9) | 5 897 (30.2) | 7 017 (35.9) | 6.0 | 6.0 | 7.0 |
| 5 | 10 648 (1.51) | 8.2 (6.8–10.2) | 3 274 (30.7) | 3 322 (31.2) | 4 052 (38.1) | 6.0 | 6.0 | 6.0 |
| 6 | 6 659 (0.94) | 8.4 (6.9–10.2) | 1 816 (27.3) | 2 204 (33.1) | 2 639 (39.6) | 6.0 | 6.0 | 5.0 |
| 7 | 4 552 (0.65) | 8.5 (7.1–10.3) | 1 132 (24.9) | 1 556 (34.2) | 1 864 (40.9) | 5.0 | 5.0 | 5.0 |
| 8 | 3 086 (0.44) | 8.6 (7.2–10.4) | 709 (23.0) | 1 058 (34.3) | 1 319 (42.7) | 5.0 | 5.0 | 5.0 |
| 9 | 1 968 (0.28) | 8.6 (7.3–10.4) | 404 (20.5) | 725 (36.8) | 839 (42.6) | 5.0 | 5.0 | 4.0 |
| 10 | 1 131 (0.16) | 8.9 (7.4–10.8) | 224 (19.8) | 367 (32.4) | 540 (47.7) | 4.0 | 4.0 | 4.0 |
| 11 | 617 (0.09) | 8.8 (7.2–10.6) | 132 (21.4) | 203 (32.9) | 282 (45.7) | 4.0 | 4.0 | 4.0 |
| ≥12 | 716 (0.10) | 9.0 (7.4–10.9) | 131 (18.3) | 236 (33.0) | 349 (33.0) | 3.0 | 3.5 | 3.0 |
| **Total** | **705 597 (100)** | | **348 962 (49.5)** | **147 741 (20.9)** | **208 894 (29.6)** | | | |

**Sequence number** IQR: Interquartile range. Seq: Sequence number

CI:1.12–1.20; p<0.001) and in those attending care at hospitals (aHR = 1.99; 95% CI:1.92–2.06; (p <0.001) (Table 5). For the 18–30 years, HbA1C improved within the first year of follow up and then was lower in longer follow-up periods compared to those over 50 years. Age group was not statistically associated with time to HbA1C control among the patients who started with out-of-control HbA1C based on the adjusted extended multivariable Cox regression model results.

## Discussion

This audit of HbA1C testing allowed us to assess diabetic control and compliance with published guidelines at both PHC facilities and hospitals in Gauteng, South Africa. We observed that poor control was present in more than 50% of those who underwent an HbA1c test (HbA1c > 7%). The majority of tests did not comply with guidelines in terms of frequency. Our study has shown that the majority of patients, even with poor control, did not have a single follow up test over four years. We noted that over 50% of all those who had an HbA1c test

**Table 4. HbA1C levels and compliance with guidelines: Proportion of patients that met testing interval guidelines according to the level of HbA1C (for first and second results).**

| Category | n (%) | 2nd HbA1C Category | | |
|---|---|---|---|---|
| | | ≤7% | Poor Control | |
| | | | >7 - ≤9% | >9% |
| Testing Interval Does Not Comply with Guidelines | 79 806 (77.2) | 31 467 (70.4) | 216 71 (82.9) | 26 668 (82.1) |
| Testing Interval Complies with Guidelines | 23 525 (22.8) | 13 235 (29.6) | 4 463 (17.1) | 5 827 (17.9) |
| **Guidelines follow-up criteria (months)** | **3–6** | **6** | **3** | **3** |
| Second HbA1C Follow-Up Median (IQR) | 11 (5–16) | 10 (5–16) | 11 (6–16) | 11 (5–17) |

Compliance is determined from first ever HbA1C test: if HbA1C <7%: follow-up in 6 months and if HbA1C>7%, follow up in 3 months

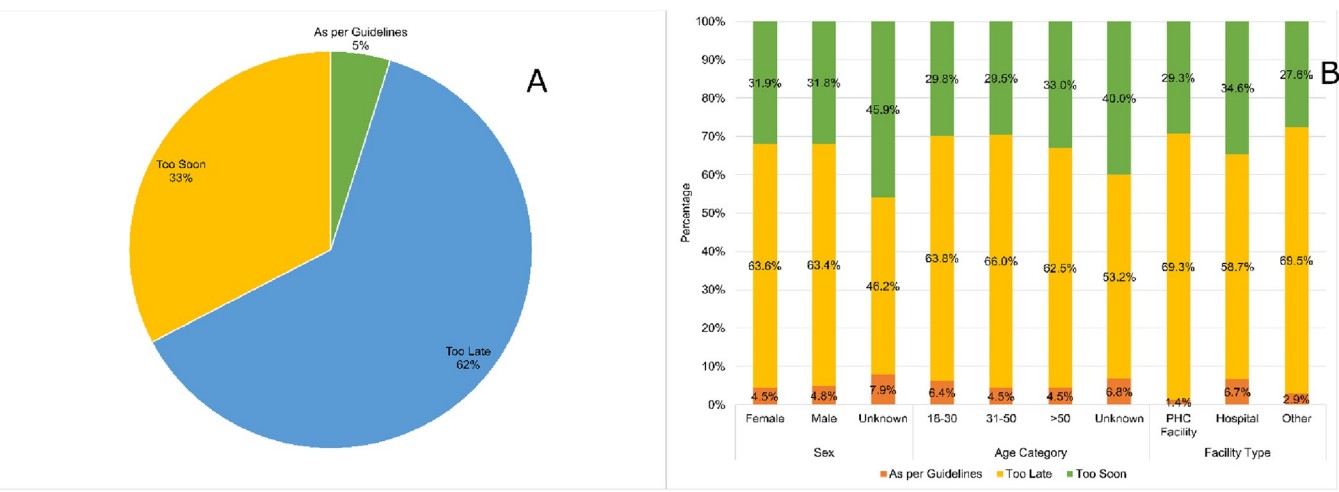

**Fig 2. Patients classification to HbA1C guidelines compliance and distribution by sex, age and facility type.** Patients classification according to HbA1C testing guidelines compliance. (A). The analysis was repeated to assess the distribution of HbA1C testing guidelines compliance by sex, age, and facility type. (B). The Y axis indicates the percentage of patients' compliance for each category.

had poor control (HbA1c > 7%). Most testing did not comply with SEMDSA guidelines in terms of frequency.

Patients under 50, Black African patients, and those attending PHC had very poor control. Hospital patients achieved better control than patients attending PHC. The possible reasons

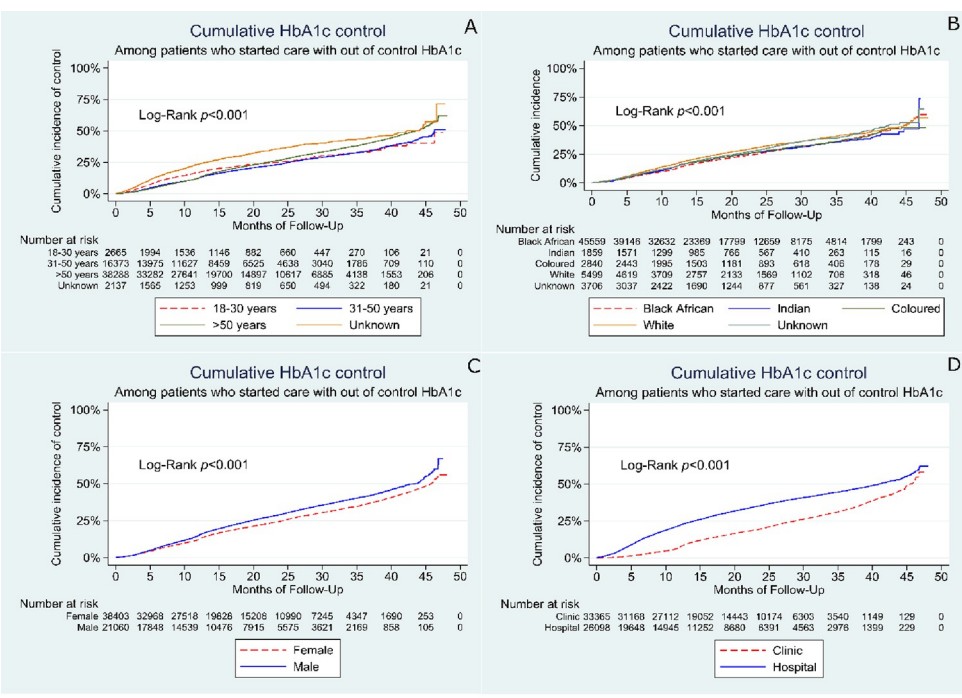

Kaplan-Meier curves showing the rate of achieving control over the period of follow-up in months by age category (A), race group (B), sex (C), and facility type (D).

**Fig 3. Rates for achieving control, describing the time to achieving HbA1c control for patients who entered into care with an out-of-control HbA1C.** Kaplan-Meier curves showing the rate of achieving control over the period of follow-up in months by age category (A), race group (B), sex (C), and facility type (D).

**Table 5. Multivariable extended Cox regression model for factors associated with time to attaining HbA1C control.**

| Characteristics | Adjusted Hazard Ratio (95% CI) | p-value |
|---|---:|---:|
| **Sex** | | |
| Female | 1 | |
| Male | 1.16 (1.12–1.20) | <0.001 |
| **Race Group (people)** | | |
| Black African | 1 | |
| Indian/Asian | 0.81 (0.73–0.88) | <0.001 |
| Coloured | 0.89 (0.82–0.96) | 0.003 |
| White | 0.99 (0.94–1.05) | 0.689 |
| Unknown | 1.06 (0.99–1.14) | 0.095 |
| **Facility Type** | | |
| PHC Facility | 1 | |
| Hospital | 1.99 (1.92–2.06) | <0.001 |
| **Age Category** | | |
| 18–30 | 1 | |
| 31–50 | 1.03 (0.94–1.14) | 0.470 |
| >50 | 1.08 (0.97–1.21) | 0.177 |
| Unknown | 0.92 (0.78–1.09) | 0.355 |

Analysis was conducted for sex, race group, facility type and age category. The adjusted hazard ratios, 95% confidence intervals and p-values are reported.

PHC: Primary Health Care

are management by diabetic specialists and access to second line therapy. Additionally, we also noted that subjects who were tested by POC, had a poorer glycaemic control than those who underwent central laboratory testing.

Previous work from the South African National Health and Nutrition Examination Survey (SANHANES-1 (2011±2012)), which looked at the diabetes care cascade has shown that there is significant loss to care [17].

The South African Diabetes guidelines as set by SEMDSA, and subsequently adopted by the Department of Health, clearly stipulate the minimum frequency of blood tests and physical examinations [10]. Forty-six percent of poorly controlled patients in this study, had a single HbA1C test for the study period, reflecting missed diagnostic opportunities for instituting changes to care such as treatment and lifestyle education, which are crucial for optimal diabetes control. It is, therefore, important to include these in the care cascade. It has been reported that reduced frequency of HbA1C testing is associated with poor control [18, 19]. For example, a United Kingdom (UK) study in diabetic patients, showed that three monthly testing was associated with a 3.8% reduction in HbA1C compared to a 1.5% increase seen with annual testing [18].

South Africa is implementing universal health coverage through a National Health Insurance scheme, and chronic diseases should be managed at the primary level. It was concerning to note that more testing occurred in hospital settings as opposed to PHCs and, while glycaemic control at both PHC and hospitals was poor, it was slightly worse in those attending PHCs. These results are similar to data reported from other urban local centres in Tshwane and Cape Town [20, 21]. This may be due to the volume of work at PHCs which have to deal with the high burden of HIV and TB as well. Task shifting to lay health workers could be an

important measure to manage increasing patient demand in PHC facilities [22]. The aHR was used to analyse factors associated with time to attain controlled HbA1C and, showed a better control in patients attending hospital. It may be because patients are referred to hospital if they are difficult to manage or have complications that require specialist attention, and benefit of second line therapy.

We were not able to assess diabetic control according to level of education and did not have data from private facilities which cater to those with medical insurance. The Global Discover Study conducted over three-years on diabetics initiating second line therapy, showed that patients from low- and middle-income countries, with no education or only primary education as well, as those attending public facilities were more likely to have poor diabetic control [23]. Patients under 50 years and Black African people are vulnerable groups that were identified in this study. Additional innovative efforts that target these groups are required such as the use of mobile clinics and screening at points where the community regularly gathers, such as queues for receiving social or pension grants. In Sweden, it was shown that people born in Africa and Asia were better reached through community-based screening than through facility-based screening [24]. The reasons for poorer control in Black Africans should be investigated. Possible reasons may be sub-optimal performance of HbA1c in this population. Ethnic differences in HbA1c concentrations have been noted in non-diabetic individuals [25, 26]. HbA1c may underestimate glycaemia in HIV positive people and to overestimate glycaemia in people with TB [27]. Poor control could also be from non-compliance with management guidelines.

In South Africa, a number of initiatives have been introduced to support the management of people with diabetes. One such intervention is the Central Chronic Medicine Dispensing and Distribution (CCMDD) programme, which provides access to medication for diabetes and other conditions, closer to home [28]. The integration of HbA1C testing into this program may be beneficial. However, it is important to note that review of the program showed that very few patients achieved targets for glycaemic control, possibly because of reduced interaction with health care workers [28]. The use of smartphone technology which has been used for HIV and TB management, can be extended to diabetes care in South Africa [29, 30]. In Guatemala, the use of community healthcare workers and smartphone technology was associated with a decline in poorly controlled diabetics over 3, 6 and 9 months [24, 31]. Other strategies could include individual provider and departmental outcome reports, patient outreach programs, patient awareness campaigns, improving electronic data systems and health care worker education. These resulted in significant improvements in both care process and clinical outcome goals in a United States of America study [32].

Contrary to our findings, several studies from other parts of the world have shown that POC testing results in improved compliance to testing guidelines, better glycaemic control, and improved patient satisfaction. Rosa et al (2021) have reported that POC HbA1C in a primary healthcare setting is a cost-effective alternative for monitoring diabetes. This study found that POC monitoring reduced the costs of diabetes-related outcomes when compared with laboratory testing [33]. Schnell et al also reported that a range of studies have demonstrated the benefits of POC HbA1C testing for improved diabetes management and glycaemic control [34]. The reasons for poor control in our setting should be determined and addressed if diabetes is to be effectively managed at primary health care centres.

Finally, the provision of national guidelines does not necessarily result in improved testing frequency as we have shown, and this is confirmed by Driskell et al [18]. There is a need for local guidance or targeted protocols based on national stakeholder involvement for development and implementation.

**Strengths and limitations of the study**

The strengths of this study lie in the large number of test results that were reviewed. The longitudinal data held in our clinical laboratory information systems provides a unique opportunity to relate testing patterns to guidelines, assess disease control or progression over time, as well as answer important clinical questions that may inspire the use of a similar strategy to monitor other chronic conditions. This study demonstrates inadequate overall testing frequency in a single province in South Africa, but also has some limitations. We were not able to distinguish between type 1 and type 2 diabetes, as well as identify pregnant women who may have undergone more frequent HbA1C testing. We were also unable to determine any treatment interventions following testing and follow up on outcomes of patients with regards to complications of diabetes. Our findings are limited to a single province and those seeking care at public facilities. A great number of individuals had only one HbA1C testing over the four-year period selected (2015–2018), and we cannot confirm if these were previously known diabetics (with good or poor control), or true first-time tested subjects.

## Conclusion

Our study is important as it confirms the findings of several other local studies that show that the majority of diabetic patients are poorly controlled, mostly younger (under 50 years), Black Africans, and that assessment of diabetic control does not meet current guidelines [17, 20, 35]. It also highlights the need to investigate the reasons for poor adherence, and to find innovative ways to improve the monitoring of diabetic patients. The broader implication of this study is that it illustrates the power of using existing laboratory data to answer important clinical questions, and because a similar strategy can be used to monitor other chronic conditions. Adherence to testing guidelines in diabetic populations is important to prevent disease complications.

## Acknowledgments

The authors would like to thank all the laboratory staff for performing HbA1C testing, and the CDW for extracting the laboratory data.

## Author Contributions

**Conceptualization:** Ngalulawa Kone, Jaya Anna George.

**Data curation:** Naseem Cassim, Innocent Maposa.

**Investigation:** Jaya Anna George.

**Methodology:** Naseem Cassim.

**Resources:** Ngalulawa Kone, Naseem Cassim.

**Supervision:** Jaya Anna George.

**Writing – original draft:** Ngalulawa Kone.

**Writing – review & editing:** Ngalulawa Kone, Jaya Anna George.

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
