## [Decision Letter · Decision Letter 0]

17 May 2023

PONE-D-22-32151Diabetic control and compliance using HbA1C testing guidelines in public healthcare facilities of Gauteng province, South Africa.PLOS ONE

Dear Dr. KONE,

Thank you for submitting your manuscript to PLOS ONE. After careful consideration, we feel that it has merit but does not fully meet PLOS ONE’s publication criteria as it currently stands. Therefore, we invite you to submit a revised version of the manuscript that addresses the points raised during the review process.

We look forward to receiving your revised manuscript.

Kind regards,

Nonhlanhla Tlotleng, PhD

Academic Editor

PLOS ONE

Journal Requirements:

Additional Editor Comments:

Dear Authors

The reviewers has suggested revisions on the manuscript. Only once the revisions has been completed will a final decision on the manuscript publication be done.

Reviewers' comments:

Reviewer's Responses to Questions

**Comments to the Author**

1. Is the manuscript technically sound, and do the data support the conclusions?

Reviewer #1: Yes

Reviewer #2: Yes

2. Has the statistical analysis been performed appropriately and rigorously? 

Reviewer #1: Yes

Reviewer #2: Yes

3. Have the authors made all data underlying the findings in their manuscript fully available?

Reviewer #1: Yes

Reviewer #2: Yes

4. Is the manuscript presented in an intelligible fashion and written in standard English?

Reviewer #1: Yes

Reviewer #2: Yes

5. Review Comments to the Author

Reviewer #1: This study assesses diabetic control and compliance of testing guidelines using HbA1c and factors associated with time to achieve control in South Africa.

Page 5: Reference 3 is incorrectly referenced.

Page 8: PHC in full when used for the first time.

Figure 1 and page 10: The exclusion (inclusion) criteria included an age of ≥ 18 years.

Page 10: Write in full POC when used for the first time.

Tables: n (%) not n=(%)

Figure 2A: Title, Patients classification according to HbA1C testing guidelines compliance (Fig 2A) and Correlation between testing guideline with sex, age, and facility type (Fig 2B). The Y axis indicates the percentage of samples reported.

Title, Patients classification according to HbA1C testing guidelines compliance (Fig 2A) and distribution of testing guideline by sex, age category, and facility type (Fig 2B).

It is better to label the Y-axis as percentage since the X axis is already labelled.

Page 18: Table 5. The adjusted hazard ratios for Indian/Asian and Coloured are lower than for Black Africans. Can this be due to sub-optimal performance in Black African populations as reported in many studies in US and South Africa? This needs to be discussed.

The adjusted hazard ratios for Hospital is higher when compared to PHC but hospital patients performed better. This is contradictory. Verify and justify.

Discussion

The first paragraph of the discussion should present the main and novel findings before comparing them with others.

The third sentence of the first paragraph is irrelevant in that paragraph. It should constitute a paragraph with comparisons with other studies.

Page 20, paragraph 2: HbA1C test has been reported to perform sub-optimally in African populations (References from US and South African studies).

Include a section on the Strengths and limitations of the study before conclusion. The last paragraph under conclusion is on limitations but no strengths of the study. This should show the differential performance of HbA1C by ethnicity.

Reviewer #2: The article entitled “Diabetic control and compliance using HbA1C testing guidelines in public healthcare facilities of Gauteng province, South Africa” can be accepted for publication with minor corrections. The manuscript is well written and presented clearly and written in standard English.

Please address the following comments:

INTRODUCTION:

• It would be nice if you could include the following in the introduction section- the number of diabetes cases, the number of diabetic-related deaths reported, the prevalence of diabetes among different race groups and the projected number of diabetic cases in South Africa (If any latest report).

• Is there any study conducted in a developing country like SA where patients obtained sufficient screening as per their guidelines? If it is, please include.

• Use either Type II diabetes or type 2 diabetes. “Type II diabetes mellitus” is used on page 5 line 4 and “type 2 diabetes” is used on page 3 line 5

RESEARCH DESIGN, MATERIALS AND METHODS:

• The statistical analysis has been performed appropriately and rigorously

• Statistical analysis: Please include the statistical software platform that you used (SAS and Stata SE) under the heading Statistical analysis

• Page 6 line 9: change Central data warehouse (CDW) to “Central Data Warehouse (CDW)”

• Page 8 line 10: PHC full form is not given (used for the first time)

• Use either HbA1C or HbA1c. For example, is page 7 line 2 -HBA1C and lines 7, 8 and 9-HbA1c.

• Page 10 line 3: Kaplan Meier to “Kaplan-Meier”

RESULTS

• Figure 1: Exclusion criteria are given in brackets Under Met Inclusion Criteria. Please check.

• HbA1c category: The HbA1c category given on page 9 line 9 is ≤7, >7 - ≤9 and >9% and on Page 10 line 16 <7%, >7-≤9% and >9%. Please check.

• Please check the grouping categories given in Table 3

• Figure 2 is unclear

• Page 16: last sentence. Is it a clinic or PHC? Also, in Figure 3C, please check.

• Page 19 Line 17 please check if this is a correct statement “It has been reported that reduced frequency of HbA1c testing is associated with poor control”

Conclusion:

• The data support the conclusions.

• 2nd paragraph can be separated as “Limitations of the study”

6. PLOS authors have the option to publish the peer review history of their article (what does this mean?). If published, this will include your full peer review and any attached files.

Reviewer #1: No

Reviewer #2: No

---

## [Author Response · Author response to Decision Letter 0]

31 Jul 2023

Dear Nonhlanhla Tlotleng, 

Thank you for reading and advising on the changes regarding my manuscript - I really appreciate your time and effort.

Please receive my revised documents - hope they meet all the requirements to your satisfaction as well as PLOS ONE journal. I look forward to having my article published soon in PLOS ONE. Kind regards, N Kone.

---

## [Editor Report · Decision Letter 1]

3 Aug 2023

Diabetic control and compliance using HbA1C testing guidelines in public healthcare facilities of Gauteng province, South Africa.

PONE-D-22-32151R1

Dear Dr. Kone Ngalulawa

We’re pleased to inform you that your manuscript has been judged scientifically suitable for publication and will be formally accepted for publication once it meets all outstanding technical requirements.

Kind regards,

Nonhlanhla Tlotleng, PhD

Academic Editor

PLOS ONE

Additional Editor Comments (optional):

All reviewers' comments have been adequately addressed.
---

## [Editor Report · Acceptance letter]

7 Aug 2023

PONE-D-22-32151R1 

Diabetic control and compliance using glycated haemoglobin (HbA1C) testing guidelines in public healthcare facilities of Gauteng province, South Africa. 

Dear Dr. Kone:

I'm pleased to inform you that your manuscript has been deemed suitable for publication in PLOS ONE. Congratulations! Your manuscript is now with our production department. 

Kind regards, 

on behalf of

Dr. Nonhlanhla Tlotleng 

Academic Editor

PLOS ONE